# Reasons for Turnover of Kansas Public Health Officials during the COVID-19 Pandemic

**DOI:** 10.3390/ijerph192114321

**Published:** 2022-11-02

**Authors:** Cristi Cain, D. Charles Hunt, Melissa Armstrong, Vicki L. Collie-Akers, Elizabeth Ablah

**Affiliations:** 1Kansas Department of Health and Environment, 1000 SW Jackson St, Topeka, KS 66612, USA; 2Johnson County Department of Health and Environment, 11875 S Sunset Dr., Olathe, KS 66061, USA; 3Department of Population Health, University of Kansas School of Medicine-Wichita, 1010 N Kansas, Wichita, KS 67214, USA; 4Department of Population Health, University of Kansas Medical Center, 3901 Rainbow Blvd, MS 1008, Kansas City, KS 66160, USA

**Keywords:** turnover, local health departments, COVID-19, pandemic, Kansas, health officers, public health officials, administrators, burnout, county commissioners

## Abstract

Public health officials played a critical role in COVID-19 mitigation and response efforts. In Kansas, 51 local health department (LHD) administrators and/or local health officers left their positions due to the pandemic between 15 March 2020 and 31 August 2021. The purpose of this study was to identify factors that led to turnover of Kansas local public health officials during the COVID-19 pandemic. Those eligible to participate in this study included former LHD administrators and/or health officers who were employed at or contracted by a Kansas LHD on 15 March 2020 and resigned, retired, or were asked to resign prior to 31 August 2021. Researchers used a demographic survey, a focus group, and key informant interviews to collect data. Twelve former LHD leaders participated in this study. Four themes emerged from phenomenological analysis: politicization of public health; a perceived lack of support; stress and burnout; and the public health infrastructure not working. The findings of this study can guide the Kansas public health system to address the issues leading to turnover of leadership and prevent future turnover. Future research must explore strategies for mitigating leadership turnover and identify alternative public health structures that could be more effective.

## 1. Introduction

As of 31 December 2021, the COVID-19 pandemic, a public health emergency of historic magnitude, led to more than 276 million cases and 5,373,898 deaths worldwide [1]. In the United States (US), there were 52,809,291 cases, 3,699,203 hospitalizations, and 816,239 deaths [2]. Although the US has only 4.2% of the world’s population, it was responsible for 19% of the world’s COVID cases and 15% of the world’s COVID deaths.

The state of Kansas experienced 512,461 cases, 16,846 hospitalizations, and 6964 deaths related to COVID-19 from 15 March 2020 through 31 December 2021 [2]. On 16 December 2021, Kansas ranked 30th in death rate (237 per 100,000 population) and 23rd in case rate (16,964 per 100,000) (with one county representing highest rates in the country in both categories) [3,4]. In December 2020, seven Kansas counties were included in the 10 counties across the United States with the highest COVID death rates [5]. During the same period, 27 Kansas counties were included in the top 100 counties in the United States for COVID death rates [5].

Prior to the beginning of the COVID-19 pandemic, the US public health system was weakened from chronic, inadequate funding, workforce shortages, and outdated infrastructure [6]. The public health system has experienced significant shortcomings related to workforce recruitment and retention, which can largely be attributed to low pay and lack of opportunities for advancement [7]. COVID-19 exacerbated these challenges and exposed others, including gaps in informatics, politicization of public health, and the public’s mistrust of public health officials [8]. Accordingly, the reliance on outdated public health infrastructure severely hindered public health’s response to the pandemic.

Kansas has a decentralized public health system, as do 50% of states in the US, which means each county is ultimately responsible for ensuring the health and safety of their citizens [9]. The state provides funding, support, and other resources but has little oversight or authority related to local public health. Of 105 counties in Kansas, 100 have local health departments (seven counties are part of two multi-county health departments). Kansas local health departments are governed by boards of health, which are comprised of the elected county commissioners in most counties [10]. Boards of health are responsible for ensuring public health services are available in their county. In most counties, they have oversight of county health officers and local health department administrators [10]. At the beginning of the pandemic, approximately 55% of Kansas counties had administrators serving in dual roles as both administrator and health officer [11].

The COVID-19 pandemic led to implementation of public health mitigation measures that had not been implemented since the Spanish flu epidemic in 1918 [12]. In Kansas, these measures included a statewide lockdown, with only essential employees leaving their homes; school closures; mask mandates; restrictions on mass gatherings; and issuance of isolation and quarantine orders [13].

During public health’s response to COVID-19, many local public health officials in the US left their positions. The extent of this problem is unknown, as there is no central repository for this type of data, and some states do not collect the information at all. However, multiple news reports have estimated between 200 and 250 departures of local health officials across the US during the pandemic [14]. There is limited research exploring the reasons that local public health officials left their positions during the pandemic. Therefore, this study was conducted to identify the factors that led to the turnover of Kansas local public health officials during the COVID-19 pandemic.

## 2. Materials and Methods

### 2.1. Participants

Participants eligible for this study were former local health department (LHD) administrators and/or health officers who were employed at a local Kansas health department on 15 March 2020 and resigned or retired during the COVID-19 pandemic prior to 31 August 2021. Those who retired during this period and reported they would have retired at that time, regardless of the pandemic, were not included. Additionally, those who temporarily served in these roles on a short-term basis were not included.

Participants were identified by using information obtained from the Kansas Department of Health and Environment (KDHE)’s Local Public Health Program (LPHP), which is responsible for tracking information about Kansas local health department administrators and health officers. For those of whom LPHP did not have contact information, staff at Kansas local health departments were e-mailed or called, or social media (predominately Facebook) were used to access contact information. Those who met the inclusion criteria were sent an e-mail that asked them to complete a demographic survey. Through the survey, they were invited to participate in a virtual interview or focus group.

### 2.2. Instruments

Data for this project were collected through a demographic survey, a focus group, key informant interviews, and turnover data accessed through LPHP. The demographic survey, developed for this study, included 13 items assessing gender, age, race/ethnicity, level of education, rurality (e.g., frontier, rural, semi-urban, urban), geographic location, professional roles, number of years served as a public health official, number of years in public health, current employment status, and political affiliation. The survey also assessed respondents’ level of interest in participating in a virtual focus group, a virtual key informant interview, or neither. The qualitative interview script included 13 open-ended questions, including the top reasons these public health officials left their positions. The script also prompted participants to identify strategies that would support current public health officials who remained in their positions and ideas for de-politicizing public health.

### 2.3. Procedures

This project was approved by the University of Kansas Medical Center’s Institutional Review Board. To identify potentially eligible participants, the researcher contacted (via e-mail and phone) former workplaces and homes of local health officers and/or administrators who had left their positions; they were asked if their reason for retirement or departure was COVID-19 related. When identification of possible participants first began, LPHP provided a list of 44 former health department officials, 32 of whom met the inclusion criteria and were invited via e-mail to participate. Two declined participation, three could not be located, and five did not respond to the invitation.

The demographic survey was administered online via the University of Kansas Medical Center’s Research Electronic Data Capture (REDCap) account. Eligible participants’ demographics were considered to ensure the sample was representative of the intended population. Those who agreed to participate in a focus group or interview were provided sign-on information to a virtual meeting using Zoom, an online meeting platform. An informed consent statement was read prior to the focus group and each interview. Interviews and the focus group were conducted between 24 August and 16 September 2021.

### 2.4. Statistical Analysis

A phenomenological approach was used for this study to understand the experience of being a Kansas local public health official during the pandemic [15]. Qualitative data were transcribed by the researcher. An open coding procedure was used, and codes were determined after the data were reviewed. Two researchers independently coded data about why participants left their positions of authority in public health and then compared the emerging themes. When there were differences in themes, the pair discussed the rationale for including (or excluding) certain themes until agreement was reached.

### 2.5. Demographics of Participants

From 15 March 2020 through 31 August 2021, 51 Kansas local health officials (representing 41 counties) left their positions (Figure 1). It was reported that 12 of those who left were health department administrators only, 26 were health officers only, and 13 served as both administrator and health officer. Multiple counties lost more than one local health official, as denoted by numbers superimposed on counties in Figure 1. The departures occurred in all regions of the state, in frontier, rural, densely-settled rural, semi-urban, and urban communities [16]. Seventy-one percent (29 of 41) of the counties impacted were either frontier (n = 19) or rural (n = 10). In fact, only one urban county (Shawnee) experienced this turnover. Most of those who left resigned, a few retired early, two were asked to resign, and one died of COVID-19.

Of the 32 former local public health officials who met inclusion criteria, 69% (n = 22) completed the survey, and 38% (n = 12) agreed to participate in an interview or focus group. One focus group and nine interviews were conducted.

Table 1 contains descriptive information about participants. Most participants were female (83%, n = 10); therefore, only she/her pronouns were used to reference all participants to protect confidentiality of male participants. This proportion is similar to the population; 71% of Kansas local health officials were female (n = 106) in August 2020. Half (n = 6) of the participants were between 40 and 50 years of age. All participants were Caucasian/White, and none reported being Hispanic. Participants represented all regions in Kansas except for north central. Half of the participants (50%, n = 6) were from rural communities, and 25% (n = 3) were from semi-urban counties.

Five of the participants were former health department administrators, two were former health officers, and five had served in a dual role as both administrator and health officer. In Kansas, 34% (n = 51) of all local health officials served in dual roles. Half (50%, n = 6) had advanced degrees, and 33% had doctorate level degrees. Fifty-eight percent (n = 7) of participants were registered nurses. Most (75%, n = 9) participants had resigned during the pandemic, two had retired earlier than they had originally planned, and one participant was asked to resign. Two-thirds (67%, n = 8) were currently employed, 75% (n = 6) of whom remained in public health. Participants reported having worked in public health for an average of 16 years and having served in a leadership role for an average of five years. Collectively, the participants had 196 years of experience.

## 3. Results

Four themes emerged regarding the reasons that these high-level Kansas local public health officials left their positions during the pandemic: the politicization of public health during the COVID-19 pandemic; a perceived lack of support from county commissioners, other county officials, and the general public; stress and burnout; and the public health system not working. These themes included several sub-themes (Table 2).

### 3.1. Politicization of Public Health

#### 3.1.1. Extreme Political Divisiveness

Many participants commented about how public health became politicized during the pandemic and how politicians and the public were polarized. Kansas had particularly challenging state-level politics. The governor was a Democrat who imposed strict mitigation requirements which included a statewide lockdown, a statewide closure of schools, gathering size limits, and mask mandates [13]. The conservative state legislature counteracted the governor’s restrictions by passing legislation—HB 2016 and SB40—that stripped the power of the governor and local health officers that had previously enabled them to mandate mitigation measures [13].

Some participants commented about how these politics impacted public health. One expressed, “I think the reality is that the legislature really wanted to punish the governor for her actions during the pandemic, and so public health got caught in the middle”.

Some participants commented specifically about the actions of the Kansas Legislature during the pandemic. “They gave the boards [of health] the authority to overrule a decision from a local health officer. Up until that point, the board could hire and fire the health officer but couldn’t tell them what to do”.

One participant described the legislature’s actions by stating, “They need to listen to the people that have the health experience and really understand that what we’re doing is trying to protect everyone. I feel like that didn’t happen. They didn’t put the health of everybody in the forefront”.

One participant shared her thoughts about how politics have damaged public health. “My fear is that the damage done to public health from a perception perspective is so great that it’s going to take decades to re-instill trust into the public health system at all levels”.

##### Threats

Many participants described experiencing threatening behavior from the public that ranged from destroyed mailboxes to poisoned dogs and death threats. Most did not report having significant fear for their safety; however, several of them took measures to enhance safety and protection. Of those who reported receiving threats, many of the threats were delivered via social media, and several were death threats.

One participant described a specific set of incidents. “[A staff member] and I had to quarantine a difficult group. Right after we quarantined… [the staff member’s] dog got very sick. They took their dog to the veterinarian, and they said, ‘Yes, he’s been fed rat poison’”. This participant reported that her dog had also been very sick during that same period and believed he had also been fed rat poison.

Another participant described one comment on social media that was directed at her and the health department staff. “We had a Facebook comment that we should all be hauled out in the street and be lynched for lying because they didn’t believe that our first case was a true positive”.

### 3.2. Lack of Support

#### 3.2.1. Lack of Support from County Commissioners

Many participants reported that lack of support from their county commissioners (who serve as boards of health in Kansas) was one of the main reasons for their unplanned departures. In some cases, participants reported being treated poorly by commissioners in public forums. Other participants reported that commissioners did not support the decisions they were making in the interest of the public’s health and in some cases were even in direct opposition. One participant shared, “I … felt like I didn’t have any direct support… you get to the point where it’s like, ‘okay, why am I even here then?’”.

##### Lack of Understanding about Public Health

Several participants connected this lack of support to a lack knowledge of public health science and responsibilities (as boards of health) among commissioners. One participant shared,

“… when county commissioners become the county board of health, they know nothing about public health, or in a lot of cases, health in general. And on my commission, we had a farmer/rancher, a retired school administrator who substitute teaches, and a banker. No clue. They don’t even know how to read a study.”

Several participants expressed frustration that commissioners were not making decisions based on science but rather, politics. One participant noted,

“I think it became very apparent and very challenging very quickly when everything became political, and it emboldened politicians to begin to enter into the fray from a political standpoint, not a scientific standpoint. And it’s still continuing to happen today.”

Numerous participants commented about how the lack of support from their county commissions made them feel. One recalled,

“I can’t tell you how many times I went home crying, or even cried during the middle of the day while I was working. But really, most of that, if not honestly all of that, was feeling like I had absolutely no support, which really, I didn’t. But I just felt like I had none at all.”

Several respondents reported that support from commissioners would have been a key factor for them to have remained in their positions.

#### 3.2.2. Lack of Support from Other County Officials

Participants reported a lack of support from other county officials including county administrators (who serve as a direct supervisor in many cases), sheriffs, and county attorneys. One participant commented, “But without any support, you know, it felt like I was going up in a fight with absolutely no weapons. And why am I fighting my allies?”.

Another participant described the situation with two of her county officials, one of whom, the sheriff, was responsible for enforcing all isolation and quarantine orders.

“My sheriff wasn’t going to help me do quarantine and isolation at all because he felt that was infringing on civil liberties and he wasn’t going to do it. And my county administrator went behind my back and sought legal counsel and told me I was not qualified to be the health officer!”

#### 3.2.3. Lack of Support from the General Public

Nearly all participants noted a lack of support from the general public in their counties during the pandemic. Many faced widespread backlash and public opposition to their orders and recommendations. A participant described the situation as,

“…when it feels like the community that you’re trying to protect and serve is turning against you for the very things you’re trying to do for them, it really calls into question everything that you’re doing. And whether it’s worth it or not.”

One participant discussed the public’s anger directed toward public health officials. “The public health officer is the voice of response at the local level. And so that anger and uncertainty and vitriol got directed, as a result of anger towards the situation, towards the individual”.

##### Public Health Officials under the Social Media Microscope

Many participants reported facing backlash and opposition via social media. One shared, “There was a lot of name-calling on social media”. Another stated, “They had called me about every word there is”.

Several participants also discussed how commissioners who supported public health mitigation strategies faced negative consequences from their county residents. One former leader stated, “Those two commissioners that even considered voting for mask mandates were out [not re-elected]”.

##### Unrealistic Expectations, Especially from Sports Parents

Several participants discussed challenges specifically regarding children involved in sports. One stated, “Those sports parents are relentless.” Some participants, particularly those from rural and frontier counties, discussed the importance of sports in small communities. One participant from a small community stated, “When it hit the fan was when I quarantined a junior high boys basketball team. You would have thought that I quarantined the NBA!”.

Some of the participants reported that they had support from most citizens in their counties but that the supporters were not visible as it was the opposition attending commission meetings, school board meetings, or other public gatherings.

“It was tough. I will tell you that we had a lot of people in our community that silently supported us. They stopped in. They apologized for their family members. But they told us… ‘when you guys are getting absolutely lit up on social media, we can’t say anything because this is who we go to Christmas with, or this is who we do business with, or this is who we sell seed to. So we support you. Keep fighting the good fight. We cannot support you publicly.’ So that’s just small town politics.”

##### Stress and Burnout

Many participants reported experiencing extreme stress and anxiety as well as burnout and physical exhaustion due to the demands of the work, particularly at the height of the pandemic and during subsequent surges. One described the current situation at her former health department.

“They’re down like 20 people, I think… 15 to 20… and so people are doing two jobs… that’s a tough place to be…. Morale’s low because they’re just exhausted, and it’s hard to see the light at the end of the tunnel.”

Several described the heaviness associated with making frequent “life or death” decisions. One shared, “The reality is decisions that I made have the prospect of how many people may or may not get sick and how many people may or may not have the potential of dying. And that’s heavy”.

##### Working Extreme Hours

Many participants reported working significantly more hours during the pandemic. One participant stated, “It was just a nightmare, working 80 h a week”. Another described, “The pace was completely unsustainable… completely… even utilizing state contact tracers and all of those resources”. One participant shared the significant impact she was experiencing personally. “So, it was getting to a point where it was starting to affect my physical and mental health, and I wanted to stop before there would be any further damage”.

##### Perceived Pressure to Be Perfect and Accessible

Many participants reported feeling pressure from policymakers and the public to be “perfect”. Participants from small communities discussed their feelings about being closely monitored by the general public regarding their own practice of mitigation measures as well as a perceived expectation for professional perfectionism. One asserted, “It’s a lot of pressure to be constantly perfect”. One participant perceived, “Everyone was constantly watching every move I made. I had to follow every single rule to the absolute letter”.

Some participants discussed the difficulties associated with the perceived expectations that public health officials must always be available. One shared, “Burnout, just in terms of physically, I was always on call from about 5 am to 10 pm…11 pm, especially at the height of the beginning of it”.

##### Impact on Family

Several participants reported that their work during the pandemic had negatively impacted their families. One participant reported that even when she was with her family, she was not engaged with them. “I was present, but I was never present, because I was always either answering text messages, answering an email, taking a phone call or even if I was not interfacing with anybody, I was thinking…‘okay, what’s going on?’”.

##### Public Health Infrastructure Not Working

Several participants commented about the ineffectiveness of the current public health infrastructure in Kansas. One participant stated, “I think the way to support public health administrators and health officers is to take a long hard look at what boards of health at the county level in a decentralized state looks like”. Another expressed, “The county board of health, in my opinion, should not be made up of lay people that were elected to a position to make decisions on gravel roads and bridges”.

##### Poor Communication

Additionally, some participants indicated there was poor communication during the pandemic among the federal partners (Centers for Disease Control and Prevention (CDC), the White House, and the Federal Emergency Management Agency (FEMA)); the state partners including the governor’s office and the Kansas Department of Health and Environment; and local public health officials. One participant expressed frustration with the poor communication and the general response from the federal government, “If we had a better federal response, it would have looked a lot different”.

Another participant commented, “Communication [from the state] would have been great. We got destroyed a lot because things were coming out in the media before even we [at the local health department] knew”.

##### Reflecting on Decision to Leave

Most participants reported that they made the right decision by leaving their position as a local public health official. One discussed an improved quality of life. “I’ve been able to recapture some quality of life, because I certainly lost it in my last months of tenure”. Another participant shared, “I was physically and mentally having so much stress that it was unhealthy. So, I know I did the right thing”.

Although most reported that they had made a good decision for themselves to leave, several expressed undesirable feelings about their departures. One shared, “I had a lot of guilt associated with leaving when I did because it did feel like I was abandoning the ship”. Some participants lamented the change in their career trajectory. One stated, “I really thought I would retire from this position”.

##### Recommendations from Participants

Most participants offered strategies to address the root factors that led to their departures. Many discussed necessary changes to boards of health. One stated, “I’d love to see an expanded board of health that’s got community leaders who are interested in community health- and not just elected people”. Another participant suggested removing elected officials completely.

A few participants discussed the importance of providing training about public health to county commissioners, as they currently serve as boards of health. One proposed, “Remind commissioners, especially new commissioners, what the role is as the board of health”.

Some participants recommended significant changes to the Kansas public health system. One participant commented,

“One of the things I’ve learned during the pandemic is that the system doesn’t work. It didn’t work before the pandemic, and it certainly didn’t work during the pandemic. I mean, the fragmentation is absolutely not working. It’s not serving anybody. We’ve got to find a way to protect home ruling while making things more efficient and effective”.

##### Impact of Turnover

Many participants reported that the rapid turnover of Kansas public health officials during the pandemic will dramatically impact the public health system. One indicated that multiple actions including legislative changes and lack of support from county officials and the general public “will be felt for many, many years. There are centuries of collective knowledge and expertise that have been lost now. And unfortunately, there is no way to gain that back”. One voiced, “I think the public health system itself is going to reel from this pandemic and the ramifications of everything that has transpired for decades”.

## 4. Discussion

Turnover is detrimental to the public health system because it results in a loss of institutional knowledge, and it impacts the ability to provide effective services to the public [17]. From this study’s 12 participants alone, Kansas lost approximately 200 years of public health experience. When estimating the losses of all 51 former Kansas public health leaders, the loss of experience is considerable. Moreover, half of participants had left the public health system completely, which is a significant loss of expertise to the field. More than half of participants (n = 7) were 50 years or younger, which suggests a significant loss of leadership potential in the Kansas public health system.

The findings of this study suggest four factors were associated with Kansas public health officials’ turnover, one of which was the politicization of the public health response. This is consistent with other research suggesting that politics and political affiliation were associated with COVID-19 outcomes. In fact, the COVID-19 death rate is nearly three times higher in counties where at least 60% voted for Trump [18]. The death rate was even higher in counties with a higher percentage of citizens voting for Trump, and the difference was attributed to political polarization and misinformation [18]. Kansas is a conservative state. Seventy-three percent of Kansas senators are Republican, as are 67% of members of the Kansas House of Representatives [19]. More than half of Kansans (56%) voted for the ultra-conversative candidate, Donald Trump, for the 2020 presidential election [20]. In many of Kansas’s frontier and rural counties, more than 80% of Kansans voted for Trump [20], which may explain why several Kansas counties have had higher than average death rates at times during the pandemic.

In Kansas, the legislature passed bills that weakened governmental public health authority (House Bill 2016 and Senate Bill 40), as did legislatures in more than half of states that were provided draft legislation by the American Legislative Exchange Council (a corporate-backed conservative organization) [21]. Although these bills were complex, in summary, they authorized local boards of health so that they could override decisions made by local health officers. This shifted authority for public health decision-making, from relying on local public health experts to relying on local elected officials with little to no experience, education, or training in public health. Authority to close public schools shifted from public health officials to school boards [21]. Legislative action also led to the repeal of all statewide public health mandates, so all public health decisions were then required to be made at the county level, mostly by county commissioners. In essence, this resulted in the public health system having to learn about and adapt to anti-public health legislation while the under-resourced system was attempting to fight a once-in-a-century pandemic and implementing the largest vaccination campaign in United States history.

Several participants discussed interactions with county commissioners, when the commissioners did not support their decisions and, at times, were unprofessional and rude. This is consistent with DeSalvo et al.’s findings, where some elected officials were characterized as actively interfering in public health decision-making [8].

Many participants reported they had served as public health officials under conservative boards of health whose members wanted to get re-elected. As some participants reported, commissioners who supported public health measures were often voted out, and sometimes even recall elections were held for this purpose. Participants reported that boards of health made decisions based on political pressure as opposed to science. Adding to the complexity of this challenge, some participants noted that members of the public who were against mitigation measures were willing to attend commission meetings to oppose public health interventions, whereas members of the public who were supportive did not want to attend the meetings due to risk of contracting COVID because mitigation measures were not being followed and because they would have been subject to verbal attacks and possibly violence.

In the current study, most participants reported receiving threats, and some even had acts of violence carried out (e.g., poisoned dogs). One participant discussed that when her family became threatened, it significantly impacted her willingness to continue to serve. This is consistent with other Kansas data. For instance, in the Fall of 2020, local health department leaders and staff reported to the Kansas Association of Local Health Departments (KALHD) and KDHE that they had received threats from people outraged about issuance of isolation/quarantine orders and mitigation measures like mask orders and limits on mass gatherings [22].

Many participants reported being verbally attacked and threatened on social media, which was identified as a contributing factor in the development of mental health problems that some experienced. A few participants discussed how social media gave people a platform to somewhat anonymously state whatever they wanted. This is consistent with DeSalvo et al., who suggested that “social media played a prominent role in the harassment of public health officials” [8]. In fact, DeSalvo et al. suggested that elected officials encouraged and sometimes participated in the social media attacks, which undermined the response and led to further erosion of trust and credibility [8].

Several participants reported that the job had negatively impacted their physical and mental health. They discussed the impact of making life or death decisions and watching people die, when there were public health measures that could prevent illness, hospitalization, and death. Several participants reported experiences of fearing for their safety and the safety of their families. In fact, most reported perceiving that their supervisors and colleagues (e.g., county commissioners, other county officials), and even their friends and acquaintances in their communities, were against them. The results from this study are consistent with other studies of public health workers. For instance, a study of more than 26,000 public health workers across the US who worked at a health department anytime in 2020 suggested that 53% of public health workers had reported at least one symptom of depression, anxiety, or post-traumatic stress disorder (PTSD) in the previous two weeks. Thirty-seven percent reported having post-traumatic stress disorder (PTSD), 32% reported depression, 30% reported anxiety, and 8% acknowledged that they had suicidal thoughts [23]. Nearly two-thirds (59%) reported having worked more than 40 hours in a typical week since the start of the pandemic.

In a study conducted in 2020, 66% of the US public health workforce reported burnout, and 24% reported they no longer planned to remain in the public health system [24]. The possible impact of the current level of burnout among public health workers is concerning, particularly because the US public health system has lost 20% of its workforce since 2008 [24].

Several participants noted that features of and fractures in the public health system compounded challenges experienced in local communities, which may be an issue beyond Kansas. Participants clearly indicated reliance on a board of health consisting of county commissioners was problematic. Fourteen percent of states, including Kansas, designate by statute that those who serve on a local board of health are elected officials such as county commissioners [25].

In addition, as one participant noted, COVID-19 communication problems started with the federal government. The delayed federal response, inaccurate statements, and desertion of the federal pandemic playbook all contributed to the communication issues and fragmented national response [26]. Without a unified national strategy and with conflicting guidance (e.g., masks), state and local health departments were left to develop and implement a pandemic response themselves, which led to further fragmentation and impacted public health’s credibility [8].

### 4.1. Implications for Public Health in Kansas

This study offers substantial implications for public health in Kansas. As the pandemic becomes endemic, the public health workforce may not be prepared to face additional threats that were exacerbated during the height of the pandemic due to the turnover, loss of leadership, and public health efforts having nearly solely focused on the pandemic [8]. As one participant stated, “public health is at a crossroads”. It is essential to address the decades-long systemic public health problems that include inadequate funding, workforce shortages, and outdated infrastructure [27]. A final implication that cannot be understated is the future impact of the loss of public trust and credibility. Participants noted the lack of support from key players and their active undermining of public health officials’ credibility and authority. These factors diminish the ability of public health officials to mitigate COVID-19 and outbreaks of future diseases (e.g., ability to quarantine people during a measles cluster) [21].

To address some of these implications, participants made recommendations. Some of the participants discussed the need to depoliticize public health and for decisions to be based on science instead of politics. Several discussed the need for training about public health for elected officials including county commissioners and legislators. This would create greater understanding, which could lead to more educated decision-making. Further, strengthened communications approaches describing the benefit and relevance of governmental public health to county governance may be important to explore as public health systems explore ways to re-build trust and credibility.

Major systemic changes need to occur in the Kansas public health system, as discussed by some participants, because the public health system in Kansas is antiquated. Work to examine the adoption of the Foundational Public Health Services Model and to explore how to modernize governmental public health was sidelined during the pandemic and requires renewed attention. Broad and long-lasting funding and workforce development, not just support for COVID-19 surge capacity, is needed to ensure a robust public health system in the future [24]. As part of the funding shifts, it is important to ensure that funds can be used for infrastructure development (as opposed to solely categorical purposes) and that there is adequate funding to offer competitive salaries to hire high quality candidates [28]. In addition, participants noted the role of support from stakeholders as influencing their experience. This suggests that there is an opportunity to convene a group of stakeholders from across local and state public health systems to offer their public support and publicly advocate for governmental public health in a way that has not occurred previously.

In addition to practice and systems changes, this study highlights the need for additional research. This study identified root causes of public health officials leaving their roles or even public health altogether. Research is needed to explore strategies for mitigating public health leadership turnover as well as identifying alternative public health systems that might be feasible and effective in Kansas. Additional research is needed to understand how to reduce divisiveness and how to promote public health without a political tone, to regain credibility and trust of the public. Due to the conservative culture in most Kansas counties, it would be of particular importance to have research about how to promote public health in this climate.

### 4.2. Limitations

There are some possible limitations to this study, including the sample size (n = 12), which was 24% of the population (n = 51). However, studies of relatively homogeneous populations with focused objectives commonly reach saturation after relatively few interviews (n = 9–17) or focus groups (n = 4–8) [29]. Participants’ responses were very consistent among interviewees and focus group participants. Accordingly, results were presented with data from interviewees and focus group participants together. Saturation was reached early in the qualitative analysis process.

Additionally, the study may suffer from self-selection bias [30]. Participation in the study was voluntary, and participants self-selected to become involved; accordingly, they may not have been representative of the entire group of public health officials who left their positions. However, data from the most recent Kansas Public Health Workforce Assessment suggest that the demographics for the sample population in the study are similar to the demographics of these public health officials [31]. Another limitation was the lack of inclusion of seven former officials who were not invited to participate because KDHE’s LPHP was unaware of their departure until after the focus group and interviews were conducted.

The study data were self-reported, which can be affected by recall bias, especially since some participants recalled events that had happened several months prior to the interview/focus group, during a time when they were under significant stress, some even reporting symptoms of PTSD, which could also impact recall [32]. Moreover, participants may have been concerned about the anonymity and confidentiality of their responses, which could have impacted what information they chose to share.

Additionally, the lead author and interviewer serves in a management position at the Kansas Department of Health and Environment and works directly with local health departments. This could have impacted responses and even played a role in some potential participants’ decision to participate.

## 5. Conclusions

This study suggests that there were multiple reasons for turnover of Kansas public health leaders during the COVID-19 pandemic. Reasons included: extreme political divisiveness; threats to the public health officials and their families; lack of support from county commissioners, other county officials, and the general public; stress and burnout resulting from working extreme hours, perceived pressure to be perfect and accessible, and the fallout experienced by their families; and the public health infrastructure not working. It is important that efforts are undertaken to address these issues to prevent future turnover of public health officials.

## Figures and Tables

**Figure 1 ijerph-19-14321-f001:**
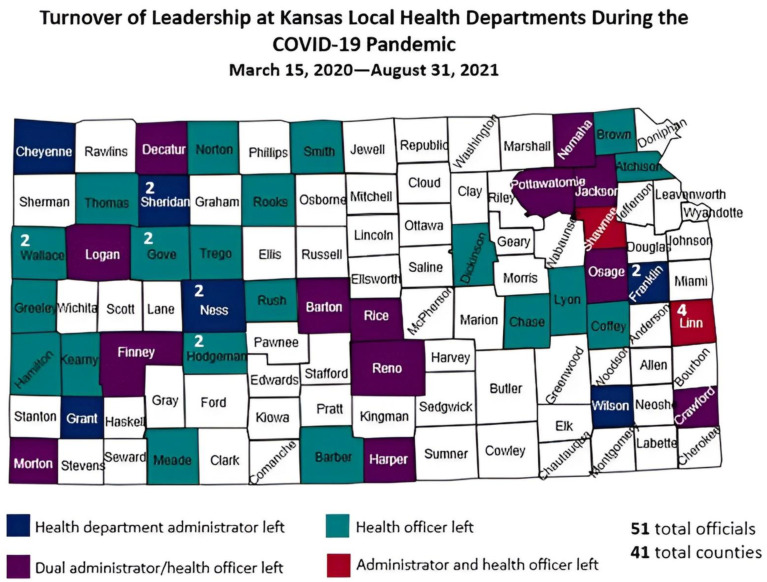
Turnover of Leadership of Kansas Local Public Health Officials during the COVID-19 Pandemic—15 March 2020–31 August 2021.

**Table 1 ijerph-19-14321-t001:** Participant Demographics and Professional Characteristics (n = 12).

Variable		Frequency	Percent
Gender	Female	10	83%
	Male	2	17%
Age (in years)	18 to 29	0	0%
	30 to 39	1	8%
	40 to 45	2	17%
	46 to 50	4	33%
	51 to 55	2	17%
	56 to 60	0	0%
	61 to 65	1	8%
	65 or older	2	17%
Race	Caucasian, White	12	100%
	African American, Black	0	0%
	Asian American, Pacific Islander	0	0%
	American Indian, Alaskan Native	0	0%
Ethnicity	Not Hispanic or Latino	12	100%
	Hispanic or Latino	0	0%
Current region of residence in Kansas	Northeast	5	42%
	Southeast	3	25%
	South Central	2	17%
	Northwest	1	8%
	Southwest	1	8%
	North Central	0	0%
	No longer live in Kansas	0	0%
County type	Rural	6	50%
	Semi-urban	3	25%
	Urban	2	17%
	Frontier	1	8%
Highest level of education	High school diploma or GED	0	0%
	Some college credit--no degree	0	0%
	Trade/technical/vocational training	0	0%
	Associate degree	3	25%
	Bachelor’s degree	3	25%
	Master’s degree	2	17%
	Professional degree	0	0%
	Doctorate degree	4	33%
Registered nurse status	Yes	7	58%
	No	5	42%
Health department leadership role	Health department administrator	5	42%
	Dual role as health department administrator and health officer	5	42%
	Health officer	2	17%
Reason for departure	Resigned	9	75%
	Retired	2	18%
	Asked to resign	1	8%
Currently employed	Yes	8	67%
	No	4	33%
Currently in public health, amongst those who were currently employed	Yes	6	75%
	No	2	25%
Current public health leadership role status	Yes	4	67%
	No	2	33%
Political affiliation	Republican	5	42%
	Independent	3	25%
	Democrat	1	8%
	Something else	0	0%
	Preferred not to answer	3	25%

**Table 2 ijerph-19-14321-t002:** Themes and Sub-themes.

Themes	Sub-Themes
Politicization of Public Health	Extreme political divisiveness Threats
Lack of Support from (1) County commissioners (Boards of Health) (2) Other county officials (3) The general public	Lack of understanding about public health Public health officials under the social media microscope Unrealistic expectations, especially from sports parents
Stress and Burnout	Working extreme hours Perceived pressure to be perfect and accessible Impact on family
Public Health Infrastructure Not Working	Poor communication

## Data Availability

The data presented in this study are available on request from the corresponding author. The data are not publicly available due to participants’ privacy.

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
