# Peer review of "Reasons for Turnover of Kansas Public Health Officials during the COVID-19 Pandemic"

_ijerph, 2022, doi:10.3390/ijerph192114321_

Round 1

Reviewer 1 Report

The topic is interesting. The results of the analyzes may have some cognitive and practical significance. The basic objection, however, is the small size of the sample. Any conclusions can only be applied to the study group. The sample size is too small to generalize the research results. In addition, the authors use two interview methods and focus groups. Conducting research using these 2 methods should be described in detail. Also, the analysis of the interviews and the results of the focus groups should be presented separately.

Author Response

Thank you for your review of and feedback about our paper, Reasons for Turnover of Kansas Public Health Officials During the COVID-19 Pandemic. We thank the reviewers for their helpful comments and time used to improve our manuscript. 

Reviewer 1: The topic is interesting. The results of the analyzes may have some cognitive and practical significance. The basic objections, however, is the small size of the sample. Any conclusions can only be applied to the study group. The sample size is too small to generalize the research results.  In addition, the authors use two interview methods and focus groups. Conducting research using these 2 methods should be described in detail.  Also, the analysis of the interviews and the results of the focus groups should be presented separately.

Response: Thank you.  Although the sample size was small, saturation was reached early in the interviews.  Studies of relatively homogeneous populations with focused objectives commonly reach saturation after relatively few interviews (n 9 – 17) or focus groups (n=4-8).  We have now added these comments to the paper.  Moreover, the results of the interviews and the focus group were very consistent.  Had responses been different, the analyses would have been presented separately.

Reviewer 2 Report

The manuscript entitled "Turnover of Kansas Public Health Officials During the COVID-19 Pandemic" aims to identify the factors that led to the turnover of Kansas local public health officials during the COVID-19 pandemic.

In my opinion, the manuscript is not yet ready for publication. Many parts need to be reviewed, such as those listed below.

Introduction

Point 1: The introduction as it is is excessively short. In this part, it is necessary to deepen the starting literature with a stronger and more recent theoretical framework from the point of view of literature. I believe it can be expanded with a more specific overview of Burnout and more recent literature that highlights the issue. It is also useful to introduce not only national but also international literature, specifically from similar countries of the study (Italy, France, Portugal, etc.). Here are some recent works that suit your theme and which I think may be useful for expanding and updating the introduction section:

- Angelini, G., Buonomo, I., Benevene, P., Consiglio, P., Romano, L., & Fiorilli, C. (2021). The Burnout Assessment Tool (BAT): A contribution to Italian validation with teachers’. Sustainability13(16), 9065.

- Buonomo, I., Santoro, P. E., Benevene, P., Borrelli, I., Angelini, G., Fiorilli, C., ... & Moscato, U. (2022). Buffering the Effects of Burnout on Healthcare Professionals’ Health—The Mediating Role of Compassionate Relationships at Work in the COVID Era. International Journal of Environmental Research and Public Health19(15), 8966.

- Fiorilli, C., Pepe, A., Buonomo, I., & Albanese, O. (2017). At-risk teachers: the association between burnout levels and emotional appraisal processes. The Open Psychology Journal10(1).

- López-Angulo, Y., Mella-Norambuena, J., Sáez-Delgado, F., Peñuelas, S. A. P., & González, O. U. R. (2022). Association between teachers’ resilience and emotional intelligence during the COVID-19 outbreak. Revista Latinoamericana de Psicología54, 51-59.

- Pluskota, M., & Zdziarski, K. (2022). Mental resilience and professional burnout among teachers. Journal of Education, Health and Sport12(3), 249-267.

- Pyhältö, K., Pietarinen, J., Haverinen, K., Tikkanen, L., & Soini, T. (2020). Teacher burnout profiles and proactive strategies. European Journal of Psychology of Education, 1-24.

- Romano, L., Angelini, G., Consiglio, P., & Fiorilli, C. (2021). The effect of students’ perception of teachers’ emotional support on school burnout dimensions: longitudinal findings. International journal of environmental research and public health18(4), 1922.

- Rusu, P. P., & Colomeischi, A. A. (2020). Positivity ratio and well-being among teachers. The mediating role of work engagement. Frontiers in psychology11, 1608.

However, I invite you to deepen the literature on the subject to have a more articulated and complex overview of the variables studied.

Method

Point 2: The number of participants, the mean and standard deviation of age, and the percentage of gender are not indicated. These data must be indicated in this section.

Point 3: Pag. 5, line 50: The manuscript refers to the fact that the data were derived from anonymous quality-improvement data, and for this reason, there is no need for the approval of an ethics committee or informed consent. Typically, study subjects should be advised that their data is being used for research purposes. In this case, using data from previously administered, consent may not be required. For this reason, it should be more explicit if the search falls within the realm of "compatible use". Furthermore, since these are published data, I believe it is necessary to consult your Internal Review Board (IRB) for approval.

I generally believe that the whole paper needs to be fixed from the point of view of the journal template. Unfortunately, I see different fonts of different sizes, which do not respect the guidelines of the journal. The same thing also applies to the bibliography. The guidelines should be read before submitting manuscripts. I also believe that it needs to be revised again in the parts mentioned above, expanded and more cared for, both from the point of view of the layout and above all in the contents.

Author Response

Reviewer 2:  The manuscript entitled “Turnover of Kansas Public Health Officials During the COVID-19 Pandemic” aims to identify the factors that led to the turnover of Kansas local public health officials during the COVID-19 pandemic. In my opinion, the manuscript is not yet ready for publication. Many parts need to be reviewed, such as those listed below. 

Introduction

Point: The introduction as it is is excessively short. In this part, it is necessary to deepen the starting literature with a stronger and more recent theoretical framework from the point of view of literature.

Response: Thank you for dedicating time to improving our paper, especially for providing these references! These are terrific. The research that has been dedicated to understanding burnout among various professionals is extremely important. Unfortunately, seven of the eight references provided address burnout among teachers, and the remaining reference addresses healthcare professionals. Although some healthcare professionals’ COVID-19 experiences may be similar to experiences of health department professionals, they are distinctly different. This is another reason why the current study is extremely important to publish and disseminate.

Also, we agree that a theoretical framework would be informative, but the study was not designed to use or test a theoretical framework. However, the publication of our study could be used to launch additional research using and/or testing various theoretical frameworks.

Reviewer 2: I believe it can be expanded with a more specific overview of Burnout and more recent literature that highlights the issue, specifically from similar countries of the study (Italy, France, Portugal, etc.). Here are some recent works that suit your theme and which I think may be useful for expanding and updating the introduction section. However, I invite you to deepen the literature on the subject to have a more articulated and complex overview of the variables studied.

[Eight references presented]

Response: There is extremely limited research that has been conducted with local health departments, let alone with the leaders of these departments in the United States during COVID-19. Our study was conducted as an exploratory study to identify the reasons that led to the turnover of these leaders; we did not know what the reasons for turnover would be. Had we been conducting a confirmatory analysis, we would have absolutely outlined and described the variables we were testing.

Reviewer 2:  Method

Point 2: The number of participants, the mean and standard deviation of age, and the percentage of gender are not indicated. These data must be indicated in this section.

Response: Thank you! The information you have highlighted is important, and we have presented it in the results section. Had we set out to include a specific proportion of participants based on age or gender, we would have included this information in our methods section instead.

Reviewer 2: Point 3: Pag. 5, line 50: The manuscript refers to the fact that the data were derived from anonymous quality-improvement data, and for this reason, there is no need for the approval of an ethics committee or informed consent. Typically, study subjects should be advised that their data is being used for research purposes. In this case, using data from previously administered, consent may not be required. For this reason, it should be more explicit if the search falls within the realm of “compatible use.” Furthermore, since these are published data, I believe it is necessary to consult your Internal Review Board (IRB) for approval.

Response: Thank you for your comments, and we appreciate the opportunity to clarify! This was not a quality improvement project, and the project was approved by the Institutional Review Board at the University of Kansas Medical Center. In fact, the data were not from a previously administered data collection; they were collected for this research in particular. We are not finding the reference to anonymous quality-improvement data on page 5, line 50 or elsewhere in the manuscript, but we would appreciate the opportunity to rectify any content that is incorrect!

Reviewer 2: Point 3: I generally believe that the whole paper needs to be fixed from the point of view of the journal template. Unfortunately, I see different fonts of different sizes, which do not respect the guidelines of the journal. The same thing also applied to the bibliography. The guidelines should be read before submitting manuscripts. I also believe that it needs to be revised again in the parts mentioned above, expanded and more cared for, both from the point of view of the layout and above all in the contents.

Response: We appreciate your careful review of our paper. We have tightened the content, updated fonts and sizes, and re-aligned the paper’s format with journal’s guidelines. We identified one area (Table 1) that used Arial point 11 and the Running head had used a different font, instead of Arial point 12 font. If we have missed other differences, we would appreciate the opportunity to rectify the inconsistent formatting! Thank you!

Reviewer 3 Report

I would like to thank the authors for this research that aims to identify the factors that led to the turnover of Kansas local public health officials during the COVID-19 pandemic.

The research subject is timely, innovative, and highly interesting. It also fits the aim and scope of the journal.

The research is well designed and follows a sound scientific research method. 

Results are clear and could have an impact among the community of researchers.

Conclusion, implications and limitations are clearly described.

However, the research needs minor adjustments:

The title needs adjustment:

We can talk about Reasons of turn over or how to justify turnover of….

Lines 400-402: You said:” In many of Kansas’s frontier and rural counties, more than 80% of Kansans voted for Trump (Politico, 2020), which may explain why several Kansas counties have had higher than average death rates at times during the pandemic”. Could you better explain the link between the vote for Trump and higher death rates?

Lines 410-411: You said:” This shifted authority for public health decision-making, from relying on public health experts to relying on elected officials with little to no experience, education, or training in public health.  Do you refer to Trump and his party members?” I agree, but it would be better to support by some examples about how politicians influenced the decision-making process.

You can diversify your sources and solidify your finding by recent references.

Other minor comments are directly attached to the manuscript.

Author Response

Reviewer 3: I would like to thank the authors for this research that aims to identify the factors that led to the turnover of Kansas local public health officials during the COVID-19 pandemic. The research subject is timely, innovative, and highly interesting. It also fits the aim and scope of the journal. The research is well designed and follows a sound scientific research method.

Results are clear and could have an impact among the community of researchers. Conclusion, implications and limitations are clearly described.

Response: Thank you!

Reviewer 3: However, the research needs minor adjustments:

The title needs adjustment:

We can talk abut Reasons of turn over or how to justify turnover of…

Response: Thank you for the suggestion. We have added “Reasons for…” to the beginning of the title.

Reviewer 3: Lines 400-402: You said, “In many of Kansas’s frontier and rural counties, more than 80% of Kansans voted for Trump (Politico, 2020), which may explain why several Kansas counties have had higher than average death rates at times during the pandemic.” Could you better explain the link between the vote from Trump and higher death rates?

Response: Thank you for your question. We added some clarifying language about the increased death rates and the reasons the authors of the study attributed to the difference – political polarization and misinformation.

Reviewer 3: Lines 410-411: You said, “This shifted authority for public health decision-making, from relying on public health experts to relying on elected officials with little to no experience, education, or training in public health. Do you refer to Trump and his party members?” I agree, but it would be better to support by some examples about how politicians influenced the decision-making process.

Response: Thank you for asking. Although an argument can be made to refer to the Trump administration’s lack of experience or education in public health, this was referring to local public health decision-making, as supported by the data presented in the results section (especially in the sub-theme of lack of understanding about public health, pages 13-15). We have edited the manuscript to more accurately reflect our intent.

Reviewer 3: You can diversify your sources and solidify your finding by recent references.

Response: Thank you for the recommendations! The Messabia et al. 2022 article addresses the management of restaurants during COVID-19, whereas our content addresses the loss of public health leaders. Regardless, your point is a good one, and we have added to and diversified our sources.

Reviewer 3: Other minor comments are directly attached to the manuscript.

Response: Thank you! We have made appropriate edits based on your comments.

Round 2

Reviewer 2 Report

Thank you for your answers, which are always an important moment of exchange. However, I do not agree with some points.

The references indicated also refer to teachers and health professionals as helping professions and I believe they could have increased the depth of the introduction. They were also an invitation to look for further ideas: in fact, I believe that 30 bibliographic entries are few for a paper on this journal.

I also believe it is important to present the characteristics of the participants (number, mean, standard deviation of age, and the percentage of gender) already in the method part. Furthermore, this aspect is present in many IJERPH papers. Perhaps it is necessary to look at the guidelines better.

Reference to approval by the Institutional Review Board at the University of Kansas Medical Center must be explicit in the text, not just in the reviewer's response.

I really appreciate your efforts to improve the paper, but I believe it is possible to review these aspects for such a relevant journal.

Author Response

Reviewer #2: The references indicated also refer to teachers and health professionals as helping professions and I believe they could have increased the depth of the introduction. They were also an invitation to look for further ideas: in fact, I believe that 30 bibliographic entries are few for a paper on this journal.

Response: We appreciate your comments. During the previous revision, we added six bibliographic entries. In examining similar papers published by the journal, we found that the number we have included (32) is consistent, especially for studies where there was extremely limited research about the population. 

Reviewer #2: I also believe it is important to present the characteristics of the participants (number, mean, standard deviation of age, and the percentage of gender) already in the method part. Furthermore, this aspect is present in many IJERPH papers. Perhaps it is necessary to look at the guidelines better.

Thank you for your guidance. We have moved the characteristics of the participants to the methods section based on your recommendation.

Reviewer #2: Reference to approval by the Institutional Review Board at the University of Kansas Medical Center must be explicit in the text, not just in the reviewer's response.

Response: Text regarding IRB approval is included in the manuscript. Please see page 7, lines 142-143 and page 30, 625-626. 
